# Performance of Urine Reagent Strips in Detecting the Presence and Estimating the Prevalence and Intensity of *Schistosoma haematobium* Infection

**DOI:** 10.3390/microorganisms10102062

**Published:** 2022-10-19

**Authors:** Abraham Degarege, Abebe Animut, Yohannes Negash, Berhanu Erko

**Affiliations:** 1Department of Epidemiology, College of Public Health, University of Nebraska Medical Center, Omaha, NE 68198, USA; 2Aklilu Lemma Institute of Pathobiology, Addis Ababa University, Addis Ababa P.O. Box 1176, Ethiopia

**Keywords:** urine reagent strips, urine filtration, *Schistosoma haematobium*, school children, Ethiopia

## Abstract

The performance of the urine reagent strips (URS) in detecting the presence and estimating the intensity of *Schistosoma haematobium* infection was evaluated using urine filtration microscopy as a reference standard. Urine samples collected from 1288 school-age children living in five villages of the Afar and one village in the Gambella Regional States of Ethiopia between October 2021 and April 2022 were examined using urine filtration and URS. The prevalence of *S. haematobium* infection was 31.6% based on urine filtration and 32.1% using URS. Using results of the urine filtration as a reference, the sensitivity, specificity, negative predictive values, and accuracy of the URS in detecting *S. haematobium* egg-positive urine specimens were 73.7%, 87.8%, 87.1%, and 82.8%, respectively. Sensitivity increased significantly with an increase in the urine egg count. Specificity was greater in low prevalence settings and among children aged 5–9 years. The level of hematuria detected was trace (19.1%), weak (30.2%), moderate (36.0%), or high (14.7%). The log odds of showing higher-level hematuria significantly increased as the number of egg counts in urine increased. In conclusion, URS remains good in rapidly screening individuals for *S. haematobium* infection, but the sensitivity of the test could be lower, particularly when the intensity of the infection is light.

## 1. Introduction

Human schistosomiasis is one of the neglected tropical parasitic diseases caused by blood flukes of the genus *Schistosoma*. Six *Schistosoma* species can infect humans, namely *Schistosoma mansoni, S. haematobium, S. japonicum, S. mekongi, S. intercalatum,* or *S. guineensis* [1]. Except for *S. haematobium,* which causes urogenital schistosomiasis, the rest of the species cause intestinal schistosomiasis. Despite this diversity in schistosome species, most schistosomiasis cases are due to *S. mansoni* and *S. haematobium.* Geographically, about 90% of human schistosomiasis cases and an estimated 280,000 deaths occur in sub-Saharan Africa [2]. *S. haematobium* affects more than 112 million people in the world, mainly in sub-Saharan Africa and the Middle East, causing urogenital schistosomiasis [3,4].

In Ethiopia, urogenital schistosomiasis is endemic in the lowlands of Afar, Gambella, Beneshangul Gumuz, and the Somali Regional States, with varying degrees of infection [5,6]. The prevalence of *S. haematobium* infection reported among children was 35.9% in those living in Abobo primary school in Gambella Regional State [5], 20.8% in those living in seven villages of Afar Regional State [7], and 47.6% in those living in Hassoba [8]. Another study also reported *S.*
*haematobium* with a prevalence of 24.2% among children in Abobo village of Gambella Regional State, 5.6% among children in Kumruk village of Beneshangul Gumuz Regional State, and 7.0% among children in Hassoba-Buri village of Afar Regional State [9]. Despite the ongoing preventive treatment of school-age children using praziquantel since 2015 [10], studies done so far reveal that urogenital schistosomiasis remains a public health threat in at least the three regional states of the country [5,6,7,8,9].

Periodic mass drug administration with praziquantel in school-age children to control schistosomiasis is being implemented upon the recommendation by WHO in endemic countries, including Ethiopia. However, in sub-Saharan Africa, children in endemic areas get infected by two years of age, and most remain with chronic infections through their school age and adulthood. As schistosome-infected preschool children are not routinely screened and treated, they constitute a potentially high-risk group for accumulation of morbidity partly because of a lack of child-friendly pediatric praziquantel formulation [11,12].

Mass treatment with praziquantel is expected to change the dynamics of *S. haematobium* infection, which is measured using diagnostics at the individual, population, and environmental levels [13]. To quantify the burden of and contain the transmission of *S. haematobium*, the availability of accurate diagnostics remains a top priority. However, there is no gold standard approach universally recommended for diagnosing *S. haematobium* infection. Although counting the parasite eggs in urine under the microscope has been a standard approach [3,4], the accuracy of the test may vary with the intensity and prevalence of infection, the number of days the sample was collected, and the filtration technique [3,4]. Due to the ongoing control programs that could lead to reduced prevalence and intensity of infection in endemic regions, the sensitivity of urine filtration microscopy may decrease significantly. Diagnosis of samples collected over multiple days may improve the sensitivity of the urine filtration microscopy [2,3,4], but this will incur cost and time.

In the effort to have the best diagnostics for *S. haematobium* infection, there have been evaluations of tools for the detection of hematuria (urine reagent strips), antigens (circulating anodic or cathodic antigen) or DNA (molecular tests) in urine or antibody released against the parasite in the blood (ELISA) [2,3,4]. Detection of hematuria using urine reagent strips (URS) has been widely used as a proxy for *S. haematobium* infection. As URS are less expensive, friendly, return results very quickly, and are less affected by the circadian production of schistosome eggs [2,3,4], they are recommended for rapid screening of *S. haematobium* infection in the field. Detection of hematuria using a reagent strip could particularly be useful for testing *S. haematobium* infection in school-aged children, as hematuria in children will strongly correlate with urogenital schistosomiasis [14]. However, the sensitivity of the URS in detecting *S. haematobium* infection showed variation in the prevalence and intensity of infection [7,9,14]. In this study, we examined the performance of the URS in detecting the presence and estimating the prevalence and intensity of *S. haematobium* infection among school-age children in selected study sites in Afar and the Gambella Regional States of Ethiopia that have been shown to have varied intensity of infection.

## 2. Methods

### 2.1. Study Area and Population

The study was undertaken in Buri, Kusra, Hassoba, Kelhat, and Andada villages in the Afar Regional State, northeastern Ethiopia, and Abobo village in the Gambella Regional State, southwestern Ethiopia (Figure 1). The villages were selected purposively. The mean annual rainfall and temperature for the villages in the Afar are 654 mm^3^ and 25.6 °C, respectively, while the corresponding values for the village in the Gambella are 68.05 mm^3^ and 32.38 °C. The majority of the inhabitants in the study areas are pastoralists who live on livestock rearing. Some inhabitants practice small-scale irrigation along the Awash River and live nearby the large-scale irrigation-based cotton farm of the Middle Awash Valley. The study villages were selected based on their endemicity for *S. haematobium* infection [7,8,9], location near the Awash River or Alwero river (Abobo district), and availability of irrigation canals and swamps. School-age children living in the selected villages from October 2021 to April 2022 were eligible for the study.

### 2.2. Study Design and Sample Size

A cross-sectional study was carried out from October 2021 to April 2022. The study was part of a project that evaluated the performance of pooled urine testing for estimating the prevalence and intensity of *S. haematobium* infection. Thus, the total sample size for both regions (*n* = 1288), specifically for each village, was estimated to address the study’s main goal of pooled urine testing.

### 2.3. Urine Sample Collection and Examination for Schistosoma haematobium Infection

Urine sample collection and examination were carried out following the procedure employed in a previous study [7]. Each child was asked to bring about 80 mL of urine in a 200 mL capacity labeled plastic container. The urine sample was collected between 10.00 a.m. and 2.00 p.m. About 10 mL of each urine sample was first tested for hematuria as a proxy for *S. haematobium* infection using Combur 10 Test reagent strips (Human GmbH-Max-Planck-Ring 21, Wiesbaden, Germany) following the manufacturer’s instructions. The results of the strip test were interpreted and recorded as zero (negative), ± (trace), + (weak), ++ (moderate), and +++ (strong) erythrocytes/μL urine or a hemoglobin concentration. Soon after the strip test, the sample was filtered and examined for *S. haematobium* eggs under a microscope on the spot. Infection intensity was determined as light (1–49 eggs/10 mL urine) or heavy (≥50 eggs/10 mL urine) using the egg count [15]. About 2 drops of 37% formaldehyde solution were added to each 10 mL of a urine sample for later processing at the Medical Parasitology laboratory of Aklilu Lemma Institute of Pathobiology, Addis Ababa University.

### 2.4. Ethical Consideration

The study obtained ethical approval from the Institutional Review Board (IRB) of the University of Nebraska Medical Center (IRB # 908-19-EP), and Aklilu Lemma Institute of Pathobiology at Addis Ababa University (Ref. No. ALIPB-IRB/10/2012/20). Permission to carry out the study in the villages was also obtained from the regional, zonal, and district health bureaus and village administrations. Written consent was obtained from children’s parents/guardians after the purpose of the study was explained to them. Only children who assented participated in this study. Children found positive for *S. haematobium* were treated with praziquantel free of charge (40 mg/kg body weight).

### 2.5. Data Analysis

Data were analyzed using Stata (StataCorp, College Station, TX, USA, v 16). A negative binomial zero-inflated regression model was used to test differences in mean egg count between age groups, gender, villages, and regions where children were living. Chi-square was used to test the binary association of the prevalence of *S. haematobium* infection with age groups, gender, villages, and regions where children were living. A multivariable logistic regression analysis was used to test the association of demographic factors with *S. haematobium* infection. The performance of the URS was assessed by calculating the sensitivity, specificity, and negative predictive values, accuracy, and the receiver operating characteristic (ROC) curve using the results of urine filtration as a gold standard. The Kappa test was used to assess the agreement between the URS and urine filtration in detecting the presence and estimating the intensity of infection. Agreement between the two tests was defined as poor (<0.20), fair (0.21–0.40), moderate (0.41–0.60), substantial (0.61–0.80), and perfect (0.81–1.00) based on the kappa values [16]. Ordered regression analysis was used to predict the effect of the egg count in urine on the level of hematuria (negative, trace, weak, moderate, and heavy). Ordered regression analysis was also used to test the relationship between classes of the intensity of infection (negative, light, heavy) and the level of hematuria (negative, trace, weak, moderate, and heavy). *p*-values were considered significant when less than 0.05.

## 3. Results

### 3.1. Prevalence of S. haematobium Infection Using Urine Filtration and Urine Reagent Strips

The demographic characteristics of the study participants and the corresponding prevalence of *S. haematobium* infections based on urine filtration and the URS methods are summarized in Table 1. Urine samples were collected from 1288 school-age children (mean of age in years ± standard deviation = 9.78 ± 3.45 years; range: 5 to 15 years) living in the Afar (*n* = 810) and Gambella (*n* = 478) regions. Children in the Afar region were from Buri, Kusra, Hassoba, Kelhat, and Andada villages, while those in the Gambella region were from the Abobo village. A greater proportion (58.3%) of the children were males. The proportions of children aged 5 to 9 years (51.6%) and 10 to 15 years (48.4%) were similar.

The prevalence of *S. haematobium* infection among the children was 31.6% on the basis of the urine filtration method and 32.1% using URS. The prevalence of infection determined using the urine filtration method was significantly lower in children aged 5 to 9 years (27.1%) compared to those aged 10 to 15 years (36.4%) (*p* < 0.01). The odds of infection also significantly increased with an increase in the ages of the children after controlling for the effect of gender and villages in a multivariable logistic regression model (adjusted OR = 1.04, 95% CI 1.01, 1.08, *p* = 0.019). The prevalence of infection also showed significant differences across the villages (*p* < 0.01). The prevalence was the highest in Abobo village (43.7%) in the Gambella region and the lowest in Hassoba village (0.76%) in the Afar region. The odds of infection were also greater among children in the Gambella region than those in the Afar region after controlling for the effect of age and gender (adjusted OR = 2.34, 95% CI 1.82, 3.02, *p* < 0.01). However, the prevalence of infection was comparable between males (32.7%) and females (30.1%) (*p* = 0.342).

Similarly, the prevalence of infection estimated using the URS was significantly greater among children aged 10 to 15 years than those aged 5 to 9 years (*p* < 0.01) and those living in Gambella regional states than those in the Afar region (*p* < 0.01). The URS also showed the highest prevalence of infection among children living in Abobo village (40.6%) and the lowest prevalence among children in Hassoba village (3.1%) (*p* < 0.01). The difference in the prevalence of infection was not significant between males and females (*p* = 0.16).

### 3.2. Intensity of S. haematobium Infection Using Urine Filtration and Urine Reagent Strips

The intensity levels of *S. haematobium* infection determined using urine filtration and URS methods are summarized in Table 2. The median of *S*. *haematobium* egg count per 10 mL of urine was 4.0 (mean = 12.89). There were 407 school-age children with eggs in their urine, of whom the great majority (95.8%) had a light-intensity infection (<50 eggs/10 mL urine) and 4.2% had a heavy-intensity infection (≥50 eggs/10 mL urine). The mean urine egg count (UEC) was significantly greater among males (16.42/10 mL urine) than in females (7.54/10 mL urine) (*p* < 0.01). The mean UEC was also greater among children living in Afar (17.18/10 mL urine) than those living in Gambella (8.82/10 mL urine) regions (*p* < 0.01). Similarly, the difference in the mean UEC estimated among children living in different villages was significant (*p* < 0.01). However, the mean UEC was comparable between children aged 10 to 15 years and those aged 5 to 9 years (3.07/10 mL urine). The difference in the distribution of the classes of the intensity of infection was significant between the Gambella and Afar regions (*p* < 0.01) and across the villages within the Afar region (*p* < 0.01).

The majority of the children who tested positive using the URS test had trace (+/−) (19.1%, 79/414), weak (+) (30.2%, 125/414), or moderate (++) (36.0%, 149/414) levels of hematuria. Only 14.7% had high (+++) levels of hematuria. In the multivariable ordered regression analysis, the log odds of showing a higher level of hematuria were greater among males than females (regression coefficient (β) = 0.26, 95% CI = 0.02, 0.49) and those living in Gambella (i.e., Abobo village) than those in the Afar region (five villages) (β = 0.54, 95% CI = 0.30, 0.78). Among children who were living in the Afar region, the log odds of showing higher-level hematuria was lower in those living in Kusra (β = −0.60, 95% CI = −0.95, −0.25), Hassoba (β = −3.04, 95% CI = −4.06, −2.02), Kelhat (β = −0.98, 95% CI = −1.83, −0.13), and Andada (β = −1.45, 95% CI = −2.42, −0.48) compared to those living in Buri. However, the difference in the log odds of showing higher-levels of hematuria was not significant between children aged 5 to 9 years and those aged 10 to 15 years (β = 0.15, 95% CI = −0.09, 0.39, *p* = 0.234).

### 3.3. Comparison of the Urine Filtration and the Urine Reagent Strips in Detecting the Presence and Estimating the Intensity of S. haematobium Infection

Table 3 summarizes results comparing the presence and intensity of *S. haematobium* infection determined using urine filtration and URS. Among the 407 children with at least one egg in their urine based on the urine filtration, 14.0% showed trace (+/−), 18.9% weak (+), 28.0% moderate (++), and 12.8% high level of hematuria by the URS. Children with heavy-intensity infections based on the urine filtration method showed weak (11.76%), moderate (52.94%), or high (35.29%) levels of hematuria via the URS. Among 390 children with light-intensity infections based on urine filtration, 14.6% showed trace, 19.2% weak, 26.9% moderate, and 11.8% heavy levels of hematuria when examine by the URS. However, 27.44% of the children with light-intensity infection using urine filtration showed no hematuria. Close to 13% (114/881) of the children with egg-negative urine using urine filtration were positive for hematuria (2.5% trace, 5.4% weak, 4.0% moderate, 1.0% high).

Assuming results of the urine filtration as a gold standard, the sensitivity, specificity, and negative predictive values of the URS in detecting *S. haematobium* egg-positive urine specimens were 73.7%, 87.8%, and 87.1%, respectively. The sensitivity of the URS in detecting egg-positive urine specimens increased significantly with an increase in the UEC, 60.3% when UEC was 1–5 per 10 mL urine, 87.0% when UEC was 6–10 per 10 mL urine, 87.9% when UEC was 11–15 per 10 mL urine, and 100% when UEC was ≥16 per 10 mL urine. The specificity (*p* = 0.015) and negative predictive values (*p* < 0.01) of the URS in detecting egg-positive urine specimens were significantly greater in Afar (88.6% and 91.9%, respectively) than in the Gambella region (83.6%, 79.2%, respectively). The performance of the URS in predicting *S. haematobium* egg-negative urine specimens was also greater in those aged 5–9 years (89.9%) than in those of 10–15 years old (85.3%) (*p* = 0.012). However, the specificity of the URS in detecting egg-negative urine specimens did not show variation by gender (female = 88.0%, male = 86.3%, *p* = 0.380) and age group (5–9 years = 87.8%, 10–15 years = 86.1%, *p* = 0.365). Similarly, the sensitivity of the URS in detecting egg-positive urine specimens did not show variation by gender (female = 71.6%, male = 75.1%, *p* = 0.160), age group (5–9 years = 73.3%, 10–15 years = 74.0%, *p* = 0.744) and the region where children live (Gambella = 71.8%, Afar = 75.8%, *p* = 0.095).

The accuracy of the URS in correctly differentiating egg-positive and egg-negative urine specimens was 82.8%. Similarly, the ROC curve, which tested all the cut points and plots the sensitivity and specificity, confirmed that the performance of URS in classifying urine samples as egg positive vs. egg negative was 82.3%. The ROC value increased to 84.5% when the effect of gender, age, and villages on UEC were controlled in the logistic regression model. The odds of detecting hematuria increased significantly with an increase in egg count after controlling for the effect of gender, age, and region where the children live in the logistic regression model (adjusted OR = 1.60; 95% CI = 1.47, 1.73). The percent agreement between urine filtration and URS in detecting the presence of *S. haematobium* egg in urine samples was 84.4% (kappa = 0.60).

The majority of the samples with weak (61.6%), moderate (76.5%), and high (85.2%) levels of hematuria using the URS also tested positive for *S. haematobium* egg using urine filtration. Among 61 children with high levels of hematuria based on URS, 10% had heavy-intensity class infections. Of the 149 children who showed moderate levels of hematuria using the URS, 6% had heavy-intensity class infections. However, only 1.6% of the children with a weak level of hematuria using the URS showed heavy intensity class infection in the urine filtration. Urine filtration test confirmed light-intensity infections among 75.4% of children with high levels of hematuria, 70.1% of children with moderate levels of hematuria, 60.0% of children with weak levels of hematuria, 72.2% of children with trace levels of hematuria, and 12.2% of children with no hematuria. The log odds of showing higher levels of hematuria significantly increased by 0.06 units as the number of eggs counted, using the urine filtration methods, increased by one (β = 0.60, 95% CI = 0.04, 0.07). Similarly, the log odds of showing higher levels of hematuria significantly increased as the classes of the intensity of infection determined using the urine egg counts increased (β = 2.89, 95% CI = 2.60, 3.19). After assigning children that showed trace, weak, and moderate levels of hematuria as light-intensity infections and those with high levels of hematuria as heavy-intensity infections, the urine filtration and URS agreed 78.4% of the times in determining the intensity class of infection as light and heavy (kappa = 0.52).

## 4. Discussion

This study evaluated the performance of the URS in detecting the presence and estimating the intensity of *S*. *haematobium* infection using urine filtration as a standard reference. URS showed excellent performance in ruling out *S. haematobium* egg-negative urine specimens (Specificity = 87.8% and Negative Predictive Value = 87.1%). However, the URS showed moderate sensitivity in accurately detecting *S. haematobium* egg-positive urine specimens (73.7%). The agreement between the URS and urine filtration was substantial in detecting egg-positive urine specimens (percent agreement = 82.4%) and determining the intensity of *S. haematobium* infection (percent agreement = 78.4%). Using the urine filtration method as a gold standard, the sensitivity and specificity of the URS in detecting egg-positive urine specimens were 73.1% and 87.7%, respectively.

A review article that combined 74 studies (102,447 participants) also estimated a 75% sensitivity and 87% specificity for detecting hematuria using URS applying a urine filtration microscope as a reference [14]. Two studies conducted in the current study area reported a higher sensitivity (>89%) of URS in detecting *S. haematobium* infections [7,9]. The decrease in the sensitivity of the URS in identifying individuals infected with *S. haematobium* in the current study could be due to the low intensity of infections that may have a negligible impact on the amount of blood in the urine, ‘hematuria’. Indeed, close to 96% of infected individuals in the current study had light intensity infection, and the probability of detecting hematuria increased with an increase in the *S. haematobium* egg count. The chance of detecting higher levels of hematuria also increased with an increase in the intensity of infection. Moreover, variation in the age range of the study subjects (school-age vs. adult), sex-specific participation, local prevalence, ongoing interventions, or treatment programs could contribute to the observed differences in the performance of the URS in detecting *S. haematobium* egg-positive urine [17].

In sum, the agreement of the URS with the urine filtration method in detecting the presence and estimating the classes of the intensity of *S. haematobium* infection was substantial (percent agreement = 84.4 and 78.4, respectively). A similar previous study also reported a similar level of agreement (Kappa = 0.64) between the URS and urine filtration in detecting the presence of *S. haematobium* infection in children [9].

The prevalence of *S. haematobium* infection was 24.4% in school-age children in the Afar region and 43.7% in the Gambella based on results from the urine filtration method. The prevalence of infection also varied significantly among the villages where the children lived (Ranges: 0.76% to 43.7%). In 2007, a study conducted in one of the study villages (Hassoba) of Afar reported a 47.6% prevalence of *S*. *haematobium* infection in children [8]. A study conducted by the current research group in the same village reported a 37% prevalence of *S*. *haematobium* infection in children in 2014 [7]. The current prevalence of infection in Hassoba (0.76%) is much lower than the previous reports [7,8]. The decrease in the prevalence of *S*. *haematobium* infections in Hassoba over the past two decades might be attributed to the drying out of marshes and ponds around the village that could serve as a habitat for the intermediate host. In addition, repeated preventive treatment in the village could have contributed to the decrease in the prevalence of infection. Moreover, the frequency of children’s contact with the swamps in Hassoba might have been reduced over time due to health education that could increase their awareness about the disease and prevention practices. Furthermore, changes in sanitation practices and safe water supply could have changed the chance of using cercariae-infested water for bathing or drinking in Hassoba.

However, the current prevalence of infection among children in Buri (34.4%) and Abobo (43.7%) was much greater than the prevalence previously reported (33.2–35.9%) [5,9]. A previous study by the team also reported a lower prevalence of infection among children in Buri compared to the current finding (24.9% and 34.4%, respectively) [7]. The increase in the prevalence of *S*. *haematobium* infection among children in Buri and Abobo might be attributed to the changes in the water bodies around the villages that serve as a habitat for the intermediate host. Children living in Abobo mainly use the water from the Alwero dam for bathing, washing clothes, and irrigation, which could increase the frequency of contact with cercariae. In addition, disparities in the availability of praziquantel drug at the health posts in Buri and Abobo and differentially less frequent mass deworming programs in these villages could have contributed to the increased prevalence of *S*. *haematobium* infections in Buri and Abobo over the past 10 years. Preschool-age children and adults living in the region who are not targets of the mass drug administration program could also serve as a source of infection, contributing to the increased prevalence of infection among school-age children in Buri and Abobo.

The prevalence of infection increased with the age of the children. Children aged at least 10 years old showed a higher prevalence of infection than those aged 5 to 9 years. Studies in Ethiopia [5], Sudan [18], Senegal [19], and Yemen [20] also documented a higher prevalence of urogenital schistosomiasis in children aged 10 to 15 years than in children aged 5 to 9 years. Due to their playing habits and socio-cultural practices, older children may engage more in water contact activities, including swimming, bathing, and fishing, which could increase the chance of exposure to cercariae. Older children may also have increased contact with swamps as they may spend more time outside helping their parents with herding, farming/irrigation activities, and watering cattle, camels, and sheep or goats. The prevalence and intensity of infection also showed significant variation in the villages where children were living. The environmental setting related to swamps and irrigation activities and children’s rate of contact with swamps infested with cercariae could also vary with villages contributing to the observed difference in the prevalence of infection across the villages.

The current results have some relevant public health implications. Although the use of urine filtration for *S. haematobium* testing requires trained personnel, power, a microscope, a large volume of urine, and a longer time for sample processing and examination, which could limit its application for monitoring and evaluation of control programs and interruption of transmission in endemic regions [21], the URS are cheap and easy to apply, provide results in short time round allowing a large number of tests in a day, and does not require power and trained personnel for use [2,3,4]. However, URS are less sensitive when the intensity of infection is low and prone to false positive results unrelated to urogenital schistosomiasis [2,3,4]. For example, different disease conditions, pregnancy, or menstrual blood could increase hematuria [22,23,24]. Hematuria associated with bladder lesions may persist even after egg excretion stops in the urine [25]. In addition, URS performance may vary with the manufacturing company [26]. Hence, URS may be cost-effective for screening large samples in a population-based survey and estimating community prevalence [27,28]. URS may also be used for individual diagnosis and treatment decisions in specific settings [29,30].

Some limitations could have affected the current results. The number of *S*. *haematobium* eggs excreted in urine might not be uniform across the days or within the time hours in a day [31]. On the other hand, urine samples in the current study were collected only on a single day and at different times. This could have increased the chance of missing lightly infected individuals, thereby underestimating the prevalence of infection in the regions. Urine filtration is less sensitive in detecting low-intensity *S. haematobium* infection [2,3,4,32]. The sensitivity and specificity of the URS could have been lower than 73% and 87%, respectively, had a more sensitive test that detect *Schistosoma* DNA, including polymerase chain reaction (PCR) been used as a reference standard [33]. Hence, the URS might be more appropriate for screening a population in moderate or high transmission and prevalence settings, but the performance of the test for detecting infection at the individual level could be lower if applied after treatment for monitoring and evaluation of urogenital schistosomiasis control programs [21]. In addition, samples were not collected from some villages in Afar and Gambella where transmission is expected to occur. Thus, the current report may not fully represent the prevalence of infection in the Afar and Gambella regions. Moreover, data on socio-economic characteristics, occupation, and water contact activities that could help better delineate infection risk factors in the regions were not collected. Furthermore, the prevalence of infection in older females might have been overestimated due to menstruation in recent days before the survey.

## 5. Conclusions

The current study confirmed that URS remains good at rapidly screening individuals for *S. haematobium* infection, but the sensitivity of the test could be lower, particularly when the intensity of infection is light. Despite differences across the villages, the prevalence of urogenital schistosomiasis among children in the Afar and Gambella regions is moderate (10–50%). Based on WHO recommendations, school-age children in Gambella and Afar could be treated in mass, employing praziquantel every two years to reduce morbidity and control the transmission of the disease in these regions [34,35]. Mass drug administration in Hassoba could be held every three years as the prevalence of infection is low (<10%) [34,35]. The risk of contracting *S. haematobium* infection may increase in males and older age children.

## Figures and Tables

**Figure 1 microorganisms-10-02062-f001:**
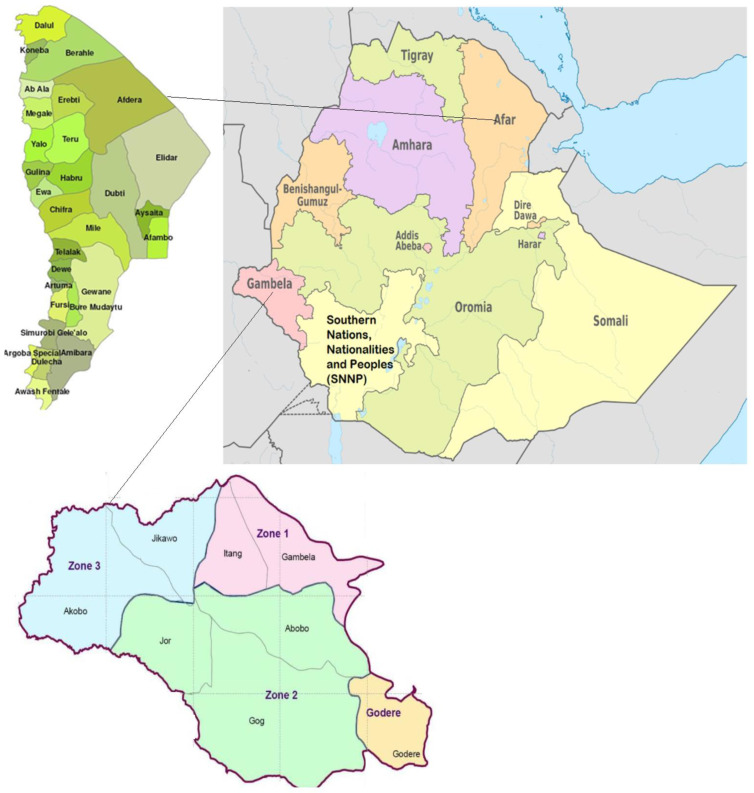
Map of the study areas in Afar and Gambella Regional States, Ethiopia.

**Table 1 microorganisms-10-02062-t001:** Prevalence of *Schistosoma haematobium* infection among school-age children in Afar and Gambella regional states of Ethiopia, October 2021 to April 2022.

Variables	Category	Number Examined	Percent Positive Based on Urine Filtration	Percent Positive Based on Urine Reagent Strips
Age in years	5 to 9	665	27.1	28.7
	10 to 15	623	36.4	35.8
	*p*-value		<0.001	0.007
Gender	Female	538	30.1	30.0
	Male	750	32.7	33.7
	*p*-value		0.342	0.155
Villages	Buri	352	34.4	38.9
	Kusra	256	25.4	26.2
	Hassoba	131	0.76	3.1
	Kelhat	34	20.6	20.6
	Andada	37	10.8	13.5
	Abobo	478	43.7	40.6
	*p*-value		<0.001	<0.001
Regions	Afar	810	24.4	27.16
	Gambella	478	43.7	40.59
	*p*-value		<0.001	<0.001

**Table 2 microorganisms-10-02062-t002:** Intensity of *Schistosoma haematobium* infection among school-age children in Afar and Gambella Regional States of Ethiopia, October 2021 to April 2022.

				Urine Filtration		Urine Reagent Strips			
Variables	Categories	Number Examined	Mean UEC	Light Infections (%)	Heavy Infections (%)	Trace (+/−)	Weak (+)	Moderate (++)	High (+++)
Age (years)	5 to 9	665	11.36	25.86	1.20	5.71	8.87	10.08	4.06
	10 to 15	623	14.10	34.99	1.44	6.58	10.59	13.16	5.46
		*p*-value	0.279	0.001		0.085			
Gender	Female	538	7.54	29.61	0.56	6.89	9.50	10.43	3.17
	Male	750	16.42	30.80	1.87	5.60	9.87	12.40	5.87
		*p*-value	<0.01	0.106		0.072			
Villages	Buri	352	15.18	32.67	1.70	5.68	11.93	14.20	7.10
	Kusra	256	23.23	25.4	1.95	3.91	10.94	7.42	3.91
	Hassoba	131	1.00	0.76	0.00	2.29	0.76	0.00	0.00
	Kelhat	34	2.57	20.59	0.00	8.82	5.88	0.00	5.88
	Andada	37	8.75	10.81	0.00	0.00	5.41	8.11	0.00
	Abobo	478	8.82	42.47	1.26 |	9.00	10.46	16.11	5.02
		*p*-value	<0.01	<0.001		<0.001			
Regions	Afar	810	17.18	23.09	1.36	4.44	9.26	8.89	4.57
	Gambella	478	8.82	42.47	1.26	9.00	10.46	16.11	5.02
		*p*-value	<0.01	<0.001		0.001			

Mean UEC (mean urine egg count) = calculated for positive samples.

**Table 3 microorganisms-10-02062-t003:** Comparison of the urine filtration microscopy and the urine reagent strips in detecting the presence and estimating the intensity of *Schistosoma haematobium* infection in Ethiopia, October 2021 to April 2022.

	Level of Hematuria Using the Urine Reagent Strips				
Egg Count per 10 mL Urine Using the Urine Filtration	Negative	Trace (+/−)	Weak (+)	Moderate (++)	High (+++)
Negative	767	22	48	35	9
1–5	93	38	47	40	16
6–10	10	13	15	24	15
11–15	4	3	4	16	6
16–30	0	2	4	19	6
31–49	0	1	5	6	3
≥50	0	0	2	9	6

Note: values within the table indicate the number of individuals with the specified test results.

## Data Availability

The data presented in this study are available on request from the corresponding author. The data are not publicly available due to privacy/ethical issues.

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
