# Peer review of "Performance of Urine Reagent Strips in Detecting the Presence and Estimating the Prevalence and Intensity of Schistosoma haematobium Infection"

_microorganisms, 2022, doi:10.3390/microorganisms10102062_

Round 1

Reviewer 1 Report

This is a solid, well-written, interesting and enjoyable paper. I certainly recommend publication.

Below I give assorted suggestions as to how it could be made more impactful to people with my interests, viz diagnostics for schistosomiasis. Basicall, I list the questions that the paper raised in my mind which you could perhaps answer for the next reader. I respectfully offer them for your consideration. 

The quality of the writing is consistently high and the paper carries the reader along. I list a few minor typos (no warranty as to completeness). 

------------------------------

Comments:

The Introduction frames the situation well. The details you provide (90 - 100) about the subjects lend urgency and immediacy to the topic and improve the flow - this is an effective touch that I have not seen before.

Just curious - how much urine did kids typically deliver, as opposed to the 150 mL asked (110)? I thought that getting more than 20 mL is sometimes a big ask.

(most important question) Can you segment the microscopy egg counts more finely? In particular, can you pull out the set with <= 5 eggs/10 mL, eg {1 - 5, 5 - 15, 15 - 50} or similar. This might allow you to hone in on the Limit of Detection of the urine strips. I believe this is highly relevant to M & E use cases where egg loads are likely to be low (Diagnostic target product profiles for monitoring, evaluation and surveillance of schistosomiasis control programmes. World Health Organization, Geneva, 670 Switzerland, 2021). So it would shed light on whether URTS are workable diagnostics for this use case.

"Ordered logistic regression" (143 ff) was a bit confusing: Did you use the URTS results as features to predict each microscopy bin, one logistic model per microscopy bin?

The mention of mean egg count (177) does not seem like the right summary statistic - what about median count over positive samples (since we know negative samples had count = 0)? Also, is this based on "exact" counts, or on the {negative, low, high} bins? If this latter, then the decimal places are maybe over-precise.

Re infection rates in males vs females: you find no significant difference (table 1), which makes the discussion (287 - 301) and (324) a bit baffling: if the news is the non-significance of difference, perhaps the focus could be more on what in the kids' environment and habits might have equalized infection rates, especially compared to prior literature (which does not invalidate the existing discussion, which is very interesting).

Figure 2 was hard for me to navigate for some reason. Could this be put into a confusion matrix? Then sensitivities and specificities at the various microscopy (i.e. urine filtration) egg counts could be put on one margin, and PPV and NPV for the different URTS readings could be put on another margin. This would be especially valuable if there was a {<=5 eggs} by microscopy bin.

Could you discuss how the URTS sens/spec vs microscopy might interact with the sens/spec vs PCR or similar (ie a really good ground truth), to give some idea about what URTS sens/spec might be vs good ground truth: is it likely to be better, the same as vs microscopy?

(Related to the above) Can you discuss how your findings relate to the performance specs given by WHO. I suppose two distinct scenarios are "test and treat" (requires accuracy per individual) and population screening (some statistical tools come into play).

Could you give a more detailed comparison of the pros and cons of URTS and microscopy, since a diagnostic must trade off several factors? eg cost, time to result, power requirements, training requirements, number of tests per day, faulty test rates, etc.

Since URTS has 88% spec vs microscopy: what non-schisto conditions might cause hematuria and thus lead to false positives (you mention menstrual bleeding). Also, is there a post-infection tail where eggs are gone but egg-induced hematuria exists (which in a population screening context might be a True Positive I suppose, depending on the length of the post-infection tail).

In the intro you mention impacts on very young children (51-55). Comment on this in the Discussion? 

The discussions re why infection rates might have changed in various regions was interesting.

--------------------------------------

Typos:

Table 2 headings: Trace, not Race

Fig 2: missing space

(46) "chemotherapy": this brings to mind radiation therapy for cancer. Is it the exact term in this case (I don't know)?

(61) are - is

(75) turn -> return

(63) show -> shows

(63-65) grammar: eg "variation in reported prevalence and intensity of infection, due to variations in sample filtration technique and number of days urine samples are collected" or similar?

(85) I don't get the use of "sensitivity" here

--------------------------------------

Author Response

Dear Dr. Manuela Ceccarelli and Ms Natalie Yan,

We would like to thank you for handling the submission and the review process of our manuscript. We would also like to thank the reviewers for their valuable and constructive comments, which helped to improve the manuscript. We made relevant revision to the manuscript on the basis of the reviewers' comments and hope that the revised manuscript meets high standards.

Below we provide point-by-point responses to the comments made by the reviewers.

Reviewer 1

This is a solid, well-written, interesting, and enjoyable paper. I certainly recommend publication. Below I give assorted suggestions as to how it could be made more impactful to people with my interests, viz diagnostics for schistosomiasis. Basically, I list the questions that the paper raised in my mind which you could perhaps answer for the next reader. I respectfully offer them for your consideration. The quality of the writing is consistently high and the paper carries the reader along. I list a few minor typos (no warranty as to completeness). The Introduction frames the situation well. The details you provide (90 - 100) about the subjects lend urgency and immediacy to the topic and improve the flow - this is an effective touch that I have not seen before.

  1. Just curious - how much urine did kids typically deliver, as opposed to the 150 mL asked (110)? I thought that getting more than 20 mL is sometimes a big ask.

Response: Thank you for catching this. The children were asked to bring 80 ml urine. The 150 ml was a type error. We have corrected that in the revised manuscript. Still, a number of children who participated in this study brought less than 80 ml. As stated in the methods section ‘study design and sample size’, this study was part of a project that evaluated the performance of pooled urine testing for estimating the prevalence and intensity of S. haematobium infection. About 80 ml urine was needed to run all the tests needed to test the hypotheses related to the pooled testing. Otherwise, 10 ml urine was enough to conduct the urine reagent test strip and urine filtration tests to generate the results in the current report (Please see line 110).

  1. (Most important question) Can you segment the microscopy egg counts more finely? In particular, can you pull out the set with <= 5 eggs/10 mL, eg {1 - 5, 5 - 15, 15 - 50} or similar. This might allow you to home in on the Limit of Detection of the urine strips. I believe this is highly relevant to M & E use cases where egg loads are likely to be low (Diagnostic target product profiles for monitoring, evaluation and surveillance of schistosomiasis control programmes. World Health Organization, Geneva, 670 Switzerland, 2021). So it would shed light on whether URTS are workable diagnostics for this use case.

Response: We thank the reviewer for this suggestion. We grouped egg count as 1 – 5 eggs/10ml urine, 6 - 10 eggs/10ml urine, 11 – 15 eggs/10ml urine, 16-30 eggs/10ml urine, 31-49 eggs/10ml urine and ≥50 eggs/10ml urine and calculated the sensitivity of the urine reagent test strips across all of these categories. Indeed, we found significant increase in the sensitivity of the urine reagent test strips as the egg count in urine increases. We have reported these results in the revised manuscript. It reads ‘sensitivity of the urine reagent test strips in detecting egg positive urines increased significantly with an increase in the UEC, 60.3% when UEC was 1-5 per 10ml urine, 87.0% when UEC was 6-10 per 10ml urine, 87.9% when UEC was 11-15 per 10ml urine, and 100% when UEC was ≥16 per 10ml urine.” (Please see lines 229 to 232).

  1. "Ordered logistic regression" (143 ff) was a bit confusing: Did you use the URTS results as features to predict each microscopy bin, one logistic model per microscopy bin?

Response: We run two ordered regression models. One model was used to predict the effect mean urine egg count on the level of hematuria. The second one was used to test relationship between classes of intensity of infection (predict the effect) and the level of hematuria. We have revised the text on the methods ‘data analyses’ to make this clear. The revised text reads as ‘Ordered regression analysis was used to predict the effect of the mean egg count in urine on the level of hematuria (negative, trace, weak, moderate and heavy). Ordered regression analysis was also used to test the relationship between classes of intensity of infection (negative, light, heavy) and the level of hematuria (negative, trace, weak, moderate and heavy) (Please see lines 144 to 147).

  1. The mention of mean egg count (177) does not seem like the right summary statistic - what about median count over positive samples (since we know negative samples had count = 0)? Also, is this based on "exact" counts, or on the {negative, low, high} bins? If this latter, then the decimal places are maybe over-precise.

Response: We have provided the median egg count estimates for only the positive samples in line 189. The revised text reads ‘The median of Shaematobium egg count per 10 ml of urine among the study participants was 4.0 (mean=12.89)” (Please see lines 186 & 187).

  1. Re infection rates in males vs females: you find no significant difference (table 1), which makes the discussion (287 - 301) and (324) a bit baffling: if the news is the non-significance of difference, perhaps the focus could be more on what in the kids' environment and habits might have equalized infection rates, especially compared to prior literature (which does not invalidate the existing discussion, which is very interesting).

Response: We have removed the text comparing sex related differences in the intensity of infection in the discussion.

The revised text/paragraph reads

“The prevalence of infection increased with the age of the children. Children with ages at least 10 years showed a higher prevalence of infection than those with ages 5 to 9 years. Studies in Ethiopia [5], the Sudan [19], Senegal [20], and Yemen [21] also documented a higher prevalence of urogenital schistosomiasis in children with ages 10 to 15 years than 5 to 9 years. Due to their playing habit and socio-cultural practices, older children may engage more in water contact activities, including swimming, bathing and fishing, which could increase the chance of exposure to cercariae. Aged children may also increase contact with swamps as they may spend more time outside helping their parents with herding, farming/irrigation activities, and watering cattle, camels, and sheep or goats. The prevalence and intensity of infection also showed significant variation in the villages where children were living. The environmental setting related to swamps and irrigation activities and children's rate of contact with swamps infested with cercariae could also vary with villages contributing to the observed difference in the prevalence of infection across the villages.” (Please see lines 339 to 351).

  1. Figure 2 was hard for me to navigate for some reason. Could this be put into a confusion matrix? Then sensitivities and specificities at the various microscopy (i.e. urine filtration) egg counts could be put on one margin, and PPV and NPV for the different URTS readings could be put on another margin. This would be especially valuable if there was a {<=5 eggs} by microscopy bin.

Response: We have replaced figure 2 with a confusion matrix table. The column of the matrix shows the Level of hematuria using the urine reagent test strips and each row indicates the ‘Egg count per 10 ml urine using the urine filtration’. The value within the cells indicates number of individuals with the specified test results. (Please see Table 3 line 275)

  1. Could you discuss how the URTS sens/spec vs microscopy might interact with the sens/spec vs PCR or similar (ie a really good ground truth), to give some idea about what URTS sens/spec might be vs good ground truth: is it likely to be better, the same as vs microscopy?

Response: We have discussed the performance of the URTS in relation to PCR. The added text in the discussion reads ‘Urine filtration is less sensitive in detecting low intensity S. haematobium infection [2-4, 23]. The sensitivity and specificity of the urine reagent test strips could have been lower than 73% and 87%, respectively had more sensitive test that detect Schistosoma DNA including polymerase chain reaction (PCR) were used a reference standard [24]. Hence, the urine reagent test strips might be more appropriate for screening a population in moderate or high transmission and prevalence settings, but the performance of the test for detecting infection at the individual level could be lower if applied after treatment for monitoring and evaluation of urogenital schistosomiasis control programs [25].’ (Please see lines 375-388)

  1. (Related to the above) Can you discuss how your findings relate to the performance specs given by WHO. I suppose two distinct scenarios are "test and treat" (requires accuracy per individual) and population screening (some statistical tools come into play).

Response: We would like to refer the reviewer to see our response above “……..Hence, the urine reagent test strips might be more appropriate for screening a population in moderate or high transmission and prevalence settings, but the performance of the test for detecting infection at the individual level could be lower if applied after treatment for monitoring and evaluation of urogenital schistosomiasis control programs [25].”

  1. Could you give a more detailed comparison of the pros and cons of URTS and microscopy, since a diagnostic must trade off several factors? eg cost, time to result, power requirements, training requirements, number of tests per day, faulty test rates, etc.

Response: We have compared the URTS with the urine filtration and discussed the public health implication of the current results (i.e URTS for diagnosing S. haematobium infection) in one paragraph in the discussion. The text reads as follows

“The current results have some relevant public health implications. While the use of urine filtration for S. haematobium testing requires trained personnel, power, a microscope, a large volume of urine, and a longer time for sample processing and examination, which could limit its application for monitoring and evaluation of control programs and inter-ruption of transmission in endemic regions [21], the URTS are cheap and easy to apply, provide results in short time round allowing a large number of tests in a day, and does not require power and trained personnel for use [2-4]. However, URTS are less sensitive when the intensity of infection is low and prone to false positive results unrelated to urogenital schistosomiasis [2-4]. For example, different disease conditions, pregnancy, or menstrual blood could increase hematuria [22-24]. Hematuria associated with bladder lesions may persist even after egg excretion stops in the urine [25]. In addition, URTS performance may vary with the manufacturing company [26]. Hence, URTS may be cost-effective for screening large samples in a population based survey and estimating community preva-lence [27,28]. URTS may also be used for individual diagnosis and treatment decisions in specific settings [29,30].” (Please see lines 354-367).

  1. Since URTS has 88% spec vs microscopy: what non-schisto conditions might cause hematuria and thus lead to false positives (you mention menstrual bleeding). Also, is there a post-infection tail where eggs are gone but egg-induced hematuria exists (which in a population screening context might be a True Positive I suppose, depending on the length of the post-infection tail).

 Response: We acknowledge that bladder lesions and associated hematuria may persist longer than the actual excretion of eggs into the bladder and have discussed that in the revised manuscript. It reads ‘Hematuria associated with bladder lesions may persist even after egg excretion stops in the urine [26].” (Please see lines 363 & 364).

  1. In the intro you mention impacts on very young children (51-55). Comment on this in the Discussion? 

Response: We have added the text below in the discussion. It reads “Preschool age children and adults living in the region who are not targets of the mass drug administration program could also serve as a source of infection, contributing to the increased prevalence of infection among school-age children in Buri and Abobo.” (Please see lines 335-338).

  1. The discussions re why infection rates might have changed in various regions was interesting.

Response: Thank you for the kind words.

  1. Typos:
  2. Table 2 headings: Trace, not Race

Response: We have replaced race with ‘trace’

  1. Fig 2: missing space

Response: We have replaced Fig 2 with Table 3

  1. (46) "chemotherapy": this brings to mind radiation therapy for cancer. Is it the exact term in this case (I don't know)?

Response: We have replaced the word chemotherapy with treatment in the revised manuscript (Please see line 47)

  1. (61) are – is

Response: We have replaced ‘are’ with ‘is’ (Please see line 63)

  1. (75) turn -> return

Response: We have replaced ‘turn’ with return’ (Please see line 75)

  1. (63) show -> shows

Response: We have replaced ‘show variation’ with ‘may vary with’

  1. (63-65) grammar: eg "variation in reported prevalence and intensity of infection, due to variations in sample filtration technique and number of days urine samples are collected" or similar?

Response: We have revised the sentence to improve the grammar. It reads as ‘Although counting the parasite eggs in urine under the microscope has been a standard approach [3,4], the accuracy of the test may vary with the intensity and prevalence of infection, the number of days the sample collected and filtration technique” (Please see line 63-65).

  1. (85) I don't get the use of "sensitivity" here

Response: We have removed the word ‘sensitivity’ from the objective. It reads ‘we examined the performance of the urine reagent test strip in detecting the presence and estimating the prevalence and intensity of S. haematobium infection among school-age children in selected study sites in Afar and the Gambella Regional States of Ethiopia that have been shown to have varied intensity of infection” (Please see lines 81-84).

Reviewer 2 Report

English language require style and minor spell check.

Author Response

Dear Dr. Manuela Ceccarelli and Ms Natalie Yan,

We would like to thank you for handling the submission and the review process of our manuscript. We would also like to thank the reviewers for their valuable and constructive comments, which helped to improve the manuscript. We made relevant revision to the manuscript on the basis of the reviewers' comments and hope that the revised manuscript meets high standards.

Below we provide point-by-point responses to the comments made by the reviewers.

Reviewer 1

This is a solid, well-written, interesting, and enjoyable paper. I certainly recommend publication. Below I give assorted suggestions as to how it could be made more impactful to people with my interests, viz diagnostics for schistosomiasis. Basically, I list the questions that the paper raised in my mind which you could perhaps answer for the next reader. I respectfully offer them for your consideration. The quality of the writing is consistently high and the paper carries the reader along. I list a few minor typos (no warranty as to completeness). The Introduction frames the situation well. The details you provide (90 - 100) about the subjects lend urgency and immediacy to the topic and improve the flow - this is an effective touch that I have not seen before.

  1. Just curious - how much urine did kids typically deliver, as opposed to the 150 mL asked (110)? I thought that getting more than 20 mL is sometimes a big ask.

Response: Thank you for catching this. The children were asked to bring 80 ml urine. The 150 ml was a type error. We have corrected that in the revised manuscript. Still, a number of children who participated in this study brought less than 80 ml. As stated in the methods section ‘study design and sample size’, this study was part of a project that evaluated the performance of pooled urine testing for estimating the prevalence and intensity of S. haematobium infection. About 80 ml urine was needed to run all the tests needed to test the hypotheses related to the pooled testing. Otherwise, 10 ml urine was enough to conduct the urine reagent test strip and urine filtration tests to generate the results in the current report (Please see line 110).

  1. (Most important question) Can you segment the microscopy egg counts more finely? In particular, can you pull out the set with <= 5 eggs/10 mL, eg {1 - 5, 5 - 15, 15 - 50} or similar. This might allow you to home in on the Limit of Detection of the urine strips. I believe this is highly relevant to M & E use cases where egg loads are likely to be low (Diagnostic target product profiles for monitoring, evaluation and surveillance of schistosomiasis control programmes. World Health Organization, Geneva, 670 Switzerland, 2021). So it would shed light on whether URTS are workable diagnostics for this use case.

Response: We thank the reviewer for this suggestion. We grouped egg count as 1 – 5 eggs/10ml urine, 6 - 10 eggs/10ml urine, 11 – 15 eggs/10ml urine, 16-30 eggs/10ml urine, 31-49 eggs/10ml urine and ≥50 eggs/10ml urine and calculated the sensitivity of the urine reagent test strips across all of these categories. Indeed, we found significant increase in the sensitivity of the urine reagent test strips as the egg count in urine increases. We have reported these results in the revised manuscript. It reads ‘sensitivity of the urine reagent test strips in detecting egg positive urines increased significantly with an increase in the UEC, 60.3% when UEC was 1-5 per 10ml urine, 87.0% when UEC was 6-10 per 10ml urine, 87.9% when UEC was 11-15 per 10ml urine, and 100% when UEC was ≥16 per 10ml urine.” (Please see lines 229 to 232).

  1. "Ordered logistic regression" (143 ff) was a bit confusing: Did you use the URTS results as features to predict each microscopy bin, one logistic model per microscopy bin?

Response: We run two ordered regression models. One model was used to predict the effect mean urine egg count on the level of hematuria. The second one was used to test relationship between classes of intensity of infection (predict the effect) and the level of hematuria. We have revised the text on the methods ‘data analyses’ to make this clear. The revised text reads as ‘Ordered regression analysis was used to predict the effect of the mean egg count in urine on the level of hematuria (negative, trace, weak, moderate and heavy). Ordered regression analysis was also used to test the relationship between classes of intensity of infection (negative, light, heavy) and the level of hematuria (negative, trace, weak, moderate and heavy) (Please see lines 144 to 147).

  1. The mention of mean egg count (177) does not seem like the right summary statistic - what about median count over positive samples (since we know negative samples had count = 0)? Also, is this based on "exact" counts, or on the {negative, low, high} bins? If this latter, then the decimal places are maybe over-precise.

Response: We have provided the median egg count estimates for only the positive samples in line 189. The revised text reads ‘The median of Shaematobium egg count per 10 ml of urine among the study participants was 4.0 (mean=12.89)” (Please see lines 186 & 187).

  1. Re infection rates in males vs females: you find no significant difference (table 1), which makes the discussion (287 - 301) and (324) a bit baffling: if the news is the non-significance of difference, perhaps the focus could be more on what in the kids' environment and habits might have equalized infection rates, especially compared to prior literature (which does not invalidate the existing discussion, which is very interesting).

Response: We have removed the text comparing sex related differences in the intensity of infection in the discussion.

The revised text/paragraph reads

“The prevalence of infection increased with the age of the children. Children with ages at least 10 years showed a higher prevalence of infection than those with ages 5 to 9 years. Studies in Ethiopia [5], the Sudan [19], Senegal [20], and Yemen [21] also documented a higher prevalence of urogenital schistosomiasis in children with ages 10 to 15 years than 5 to 9 years. Due to their playing habit and socio-cultural practices, older children may engage more in water contact activities, including swimming, bathing and fishing, which could increase the chance of exposure to cercariae. Aged children may also increase contact with swamps as they may spend more time outside helping their parents with herding, farming/irrigation activities, and watering cattle, camels, and sheep or goats. The prevalence and intensity of infection also showed significant variation in the villages where children were living. The environmental setting related to swamps and irrigation activities and children's rate of contact with swamps infested with cercariae could also vary with villages contributing to the observed difference in the prevalence of infection across the villages.” (Please see lines 339 to 351).

  1. Figure 2 was hard for me to navigate for some reason. Could this be put into a confusion matrix? Then sensitivities and specificities at the various microscopy (i.e. urine filtration) egg counts could be put on one margin, and PPV and NPV for the different URTS readings could be put on another margin. This would be especially valuable if there was a {<=5 eggs} by microscopy bin.

Response: We have replaced figure 2 with a confusion matrix table. The column of the matrix shows the Level of hematuria using the urine reagent test strips and each row indicates the ‘Egg count per 10 ml urine using the urine filtration’. The value within the cells indicates number of individuals with the specified test results. (Please see Table 3 line 275)

  1. Could you discuss how the URTS sens/spec vs microscopy might interact with the sens/spec vs PCR or similar (ie a really good ground truth), to give some idea about what URTS sens/spec might be vs good ground truth: is it likely to be better, the same as vs microscopy?

Response: We have discussed the performance of the URTS in relation to PCR. The added text in the discussion reads ‘Urine filtration is less sensitive in detecting low intensity S. haematobium infection [2-4, 23]. The sensitivity and specificity of the urine reagent test strips could have been lower than 73% and 87%, respectively had more sensitive test that detect Schistosoma DNA including polymerase chain reaction (PCR) were used a reference standard [24]. Hence, the urine reagent test strips might be more appropriate for screening a population in moderate or high transmission and prevalence settings, but the performance of the test for detecting infection at the individual level could be lower if applied after treatment for monitoring and evaluation of urogenital schistosomiasis control programs [25].’ (Please see lines 375-388)

  1. (Related to the above) Can you discuss how your findings relate to the performance specs given by WHO. I suppose two distinct scenarios are "test and treat" (requires accuracy per individual) and population screening (some statistical tools come into play).

Response: We would like to refer the reviewer to see our response above “……..Hence, the urine reagent test strips might be more appropriate for screening a population in moderate or high transmission and prevalence settings, but the performance of the test for detecting infection at the individual level could be lower if applied after treatment for monitoring and evaluation of urogenital schistosomiasis control programs [25].”

  1. Could you give a more detailed comparison of the pros and cons of URTS and microscopy, since a diagnostic must trade off several factors? eg cost, time to result, power requirements, training requirements, number of tests per day, faulty test rates, etc.

Response: We have compared the URTS with the urine filtration and discussed the public health implication of the current results (i.e URTS for diagnosing S. haematobium infection) in one paragraph in the discussion. The text reads as follows

“The current results have some relevant public health implications. While the use of urine filtration for S. haematobium testing requires trained personnel, power, a microscope, a large volume of urine, and a longer time for sample processing and examination, which could limit its application for monitoring and evaluation of control programs and inter-ruption of transmission in endemic regions [21], the URTS are cheap and easy to apply, provide results in short time round allowing a large number of tests in a day, and does not require power and trained personnel for use [2-4]. However, URTS are less sensitive when the intensity of infection is low and prone to false positive results unrelated to urogenital schistosomiasis [2-4]. For example, different disease conditions, pregnancy, or menstrual blood could increase hematuria [22-24]. Hematuria associated with bladder lesions may persist even after egg excretion stops in the urine [25]. In addition, URTS performance may vary with the manufacturing company [26]. Hence, URTS may be cost-effective for screening large samples in a population based survey and estimating community preva-lence [27,28]. URTS may also be used for individual diagnosis and treatment decisions in specific settings [29,30].” (Please see lines 354-367).

  1. Since URTS has 88% spec vs microscopy: what non-schisto conditions might cause hematuria and thus lead to false positives (you mention menstrual bleeding). Also, is there a post-infection tail where eggs are gone but egg-induced hematuria exists (which in a population screening context might be a True Positive I suppose, depending on the length of the post-infection tail).

 Response: We acknowledge that bladder lesions and associated hematuria may persist longer than the actual excretion of eggs into the bladder and have discussed that in the revised manuscript. It reads ‘Hematuria associated with bladder lesions may persist even after egg excretion stops in the urine [26].” (Please see lines 363 & 364).

  1. In the intro you mention impacts on very young children (51-55). Comment on this in the Discussion? 

Response: We have added the text below in the discussion. It reads “Preschool age children and adults living in the region who are not targets of the mass drug administration program could also serve as a source of infection, contributing to the increased prevalence of infection among school-age children in Buri and Abobo.” (Please see lines 335-338).

  1. The discussions re why infection rates might have changed in various regions was interesting.

Response: Thank you for the kind words.

  1. Typos:
  2. Table 2 headings: Trace, not Race

Response: We have replaced race with ‘trace’

  1. Fig 2: missing space

Response: We have replaced Fig 2 with Table 3

  1. (46) "chemotherapy": this brings to mind radiation therapy for cancer. Is it the exact term in this case (I don't know)?

Response: We have replaced the word chemotherapy with treatment in the revised manuscript (Please see line 47)

  1. (61) are – is

Response: We have replaced ‘are’ with ‘is’ (Please see line 63)

  1. (75) turn -> return

Response: We have replaced ‘turn’ with return’ (Please see line 75)

  1. (63) show -> shows

Response: We have replaced ‘show variation’ with ‘may vary with’

  1. (63-65) grammar: eg "variation in reported prevalence and intensity of infection, due to variations in sample filtration technique and number of days urine samples are collected" or similar?

Response: We have revised the sentence to improve the grammar. It reads as ‘Although counting the parasite eggs in urine under the microscope has been a standard approach [3,4], the accuracy of the test may vary with the intensity and prevalence of infection, the number of days the sample collected and filtration technique” (Please see line 63-65).

  1. (85) I don't get the use of "sensitivity" here

Response: We have removed the word ‘sensitivity’ from the objective. It reads ‘we examined the performance of the urine reagent test strip in detecting the presence and estimating the prevalence and intensity of S. haematobium infection among school-age children in selected study sites in Afar and the Gambella Regional States of Ethiopia that have been shown to have varied intensity of infection” (Please see lines 81-84).

Reviewer 2

Open Review

English language and style

( ) Extensive editing of English language and style required
( ) Moderate English changes required
(x) English language and style are fine/minor spell check required
( ) I don't feel qualified to judge about the English language and style

Yes

Can be improved

Must be improved

Not applicable

Does the introduction provide sufficient background and include all relevant references?

(x)

( )

( )

( )

Are all the cited references relevant to the research?

(x)

( )

( )

( )

Is the research design appropriate?

( )

(x)

( )

( )

Are the methods adequately described?

(x)

( )

( )

( )

Are the results clearly presented?

( )

(x)

( )

( )

Are the conclusions supported by the results?

( )

(x)

( )

( )

Comments and Suggestions for Authors NONE

Response: We would also like to thank the reviewer for his/her valuable and constructive comments, which helped to improve the manuscript. We made relevant revision to the manuscript on the basis of the reviewers' comments and hope that the revised manuscript meets high standards.
